# Low self-esteem in adolescents with sickle cell disease: Determinants and coping mechanisms; A mixed methods study

Janipher Nakabuye[1]*, Wani Muzeyi[1], Annet Nakirulu[2], Sarah Kiguli[1], Grace Ndeezi[1], Deogratias Munube[2]

**1** Department of Pediatrics and Child Health, Makerere University College of Health Sciences, Kampala, Uganda, **2** Department of Pediatrics and Child Health, Mulago National Referral Hospital, Kampala, Uganda

* njanipher80@gmail.com

## Abstract

### Background

Sickle Cell Disease (SCD) is a chronic genetic condition affecting approximately 112 per 100,000 live births globally, with adolescents being particularly vulnerable. Beyond its physical burden, SCD can significantly impact adolescents' mental health, particularly self-esteem. While positive coping strategies can mitigate these effects, data on self-esteem among adolescents with SCD in Uganda remain limited. The objective of this study was to determine the prevalence, associated factors, and coping mechanisms of low self-esteem among adolescents with SCD attending the Sickle Cell Clinic (SCC) at Mulago National Referral Hospital (MNRH).

### Methods

A sequential explanatory mixed-methods study was conducted at the SCC of MNRH from January to March 2024. Quantitative data were collected from 356 adolescents. Low self-esteem was defined as a total Rosenberg self-esteem score of less than 15. For analysis of factors associated with low self-esteem, modified Poisson regression with a log link and robust standard errors was used to estimate prevalence ratios (PRs) and their 95% CIs. Qualitative data on coping mechanisms were obtained through in-depth interviews with nine adolescents and three key informant interviews with health workers. Thematic analysis was done guided by the THRIVE framework.

### Results

The prevalence of low self-esteem was 21.6% (95% CI: 17.6–26.3%). Lower prevalence of low self-esteem was observed among adolescents not frequently absent from school (aPR = 0.94, 95% CI 0.90–0.99, P = 0.011) and those without limitations to physical activity (aPR = 0.94, 95% CI 0.90–0.98, P = 0.005), while limited social

**Data availability statement:** All relevant data are within the manuscript.

**Funding:** This study was funded with support from the U.S Department of State's Office of the U.S Global AIDS Coordinator and Health Diplomacy (S/GAC), and Presidents Emergency Plan for AIDS' Relief (PEPFAR) under award number 1R25TW011213. The funder had no role in study design,data collection and analysis or decision to publish or preparation of the manuscript.

**Competing interests:** The authors have declared that no competing interests exist.

engagement was associated with higher prevalence (aPR = 1.16, 95% CI 1.03–1.31, P = 0.014)."Coping mechanisms reported by adolescents included adherence to medication, healthy eating and hydration, participation in social activities and hobbies, prayer, and peer awareness.

## Conclusion

Adolescents with SCD are at considerable risk of low self-esteem, particularly those with limited physical activity, social involvement, and frequent school absenteeism. Interventions promoting healthy lifestyles, consistent medication use, peer support, and social engagement may help strengthen self-esteem in this population.

---

## 1. Introduction

Sickle cell disease (SCD) remains a significant global health burden with a birth prevalence of 112 per 100,000 live births globally and 1125 per 100,000 live births in Africa, with the highest prevalence observed in Sub-Saharan Africa, where projections indicate a continued increase by 2050 [1]. In Uganda, SCD contributes substantially to child morbidity and mortality, with an estimated 250,000–300,000 births annually, of which approximately 80% of affected children die before their fifth birthday [2]. Adolescents with SCD face unique psychosocial challenges as they transition into increased self-management of their condition. Studies have shown that SCD affects health-related quality of life, leading to increased pain episodes, frequent hospitalizations, and cognitive challenges, all of which can contribute to low self-esteem [3,4]. Evidence from previous studies suggests that adolescents with SCD may struggle with social relationships, school absenteeism, and adherence to treatment, all of which can be linked to low self-esteem [5,6]. However, some studies have reported positive self-concept among adolescents with SCD, highlighting the potential role of coping mechanisms in mitigating psychological distress and esteem [7]. While coping strategies such as spirituality, peer discussions, and engagement with mass media have been identified in other settings15]. Little is known about their role among adolescents with SCD in Uganda. With advancements in SCD care, more children in Uganda are surviving into adolescence, necessitating a greater focus on their psychological well-being. Positive self-esteem is essential for healthy psychosocial development, resilience, and adherence to disease management [5]. However, in Uganda, the prevalence of low self-esteem and the coping mechanisms used by adolescents with SCD remain unknown. The purpose of this study was to determine the burden of low self-esteem, identify associated factors, and explore coping strategies among adolescents with SCD.

## 2. Materials and methods

### Study design and setting

This was a sequential explanatory mixed-methods study conducted at the Sickle Cell Clinic (SCC) of Mulago National Referral Hospital (MNRH), Kampala, Uganda.

Quantitative data were collected using a cross-sectional design to determine the prevalence of low self-esteem and associated factors among adolescents with sickle cell disease. Subsequently, qualitative data were collected through in-depth interviews from adolescents with both normal and low self-esteem to explore their coping mechanisms. MNRH is Uganda's largest public hospital with a 1,500-bed capacity, serving as a national referral and teaching hospital affiliated with Makerere University College of Health Sciences. The SCC at MNRH receives approximately 80–120 children daily, with around 14 adolescents seen per day. The clinic is staffed by 4 hematologists, 2 medical officers, 1 clinical officer, 2 laboratory personnel, nurses, and a pharmacist. Adolescents lack a dedicated review space, and routine mental health assessments are not performed, which may affect clinic attendance.

## Sample size

A total of 356 adolescents attending the SCC at MNRH were recruited using systematic sampling, in which every second eligible adolescent was selected. This sample provided 95% power to estimate the prevalence of low self-esteem among adolescents with sickle cell disease with a precision of ±5.2% at the 95% confidence level. Adolescents who were too ill to participate in study activities were excluded.

## Study variables

### Dependent variable:
Low self-esteem, measured using the Rosenberg Self-Esteem Scale (RSES). Scores <15 indicated low self-esteem [8,9].
### Independent variables:

- *Sociodemographic:* Age, gender, caregiver relationship, religion, family size, birth order, marital status

- *Personal health:* Frequency of crises, pill burden, illness duration, mental health history

- *Socio-cultural:* School attendance, social support, home environment, recreation

- *Health system:* Access to care, quality of services, staff attitudes, medicine availability, stigma

## Statistical analysis

Data were collected using a semi-structured interviewer-administered questionnaire and the Rosenberg Self-Esteem Scale (RSES). Key predictor variables like physical activity and social engagement were measured using single-item self-report questions based on routine clinical assessments conducted in the sickle cell clinic. The RSES, a 10-item measure of global self-esteem, has been widely used across adolescent populations in sub-Saharan Africa, with such studies demonstrating Cronbach's alpha >0.6 [9,10]. No major linguistic or cultural adaptations were required beyond minor wording adjustments for clarity. All data were double-entered into EpiData version 4.4.2 to ensure accuracy and exported to STATA version 17 for analysis. Continuous variables were summarized using means with standard deviations for normally distributed data, while medians with interquartile ranges (IQRs) were used for skewed distributions. Categorical variables were summarized as frequencies and percentages. To determine the prevalence of low self-esteem, defined as a total RSES score of less than 15 [8,9], the number of adolescents meeting this criterion was divided by the total number of participants enrolled. The prevalence was reported as a percentage with corresponding 95% confidence intervals (CIs). For analysis of factors associated with low self-esteem, the outcome variable was dichotomized: 1 for "low self-esteem" and 0 for "normal self-esteem." Modified Poisson regression with a log link and robust standard errors was used to estimate prevalence ratios (PRs) and their 95% CIs. Covariate selection was data-driven, where variables with $p \le 0.20$ at bivariate analysis were considered for inclusion in the multivariable model. All eligible variables were entered into an initial multivariable model, and backward elimination was performed by sequentially removing variables with $p > 0.05$ while assessing

changes in effect estimates and overall model fit. Only variables that remained statistically significant or meaningfully improved the model were retained in the final model. Missing data were handled using complete-case analysis. Multicollinearity was assessed using variance inflation factors (VIF). Variables with VIF ≥ 10 were considered highly collinear and were excluded from the multivariable model.

### Qualitative

In-depth interviews (IDIs) were conducted with adolescents, and three key informant interviews (KIIs) were held with health workers (a nurse, medical officer, and pharmacist). Adolescents were purposively selected and randomly invited based on their RSES scores. Interviews were conducted in private settings using guides developed for IDIs and KIIs, in English or Luganda, depending on participant preference. Discussions were audio-recorded, transcribed verbatim, and translated into English. Interviews averaged 30 minutes and were moderated by a trained social scientist, with note-taking and audio recording by the PI and research assistant. Data collection continued until saturation. Transcripts were analyzed thematically using Open Code software. Transcripts were repeatedly read, and codes were developed from identified meaningful units. These codes were organized into subthemes and overarching themes to describe coping mechanisms among adolescents with SCD. Relevant quotations were used to support the thematic findings.

### Ethical considerations

Ethical approval for this study was obtained from the Makerere University School of Medicine Research and Ethics Committee and was granted Mak-SOMREC-2023–740 on 28th March 2023. Written informed consent was obtained from parents or legal guardians of adolescents under 18 years, and from adolescents aged 18–19 years. In addition, written assent was obtained from adolescents younger than 18 years.

## 3. Results

### Participant characteristics

The majority of participants (65%) were aged 10–14 years, and just over half (50.4) were female. Most (58.2%) had no formal education, and over 70% had a relative with sickle cell disease (SCD). These characteristics are summarized in Table 1.

### Prevalence of low self-esteem

Of the 356 participants, 77 (21.6%, 95% CI: 17.6–26.3%) had low self-esteem.

### Bivariate analysis of factors associated with low self-esteem

Adolescents aged >14 years were slightly less likely to have low self-esteem compared to those aged 10–14 years (CPR = 0.95, p = 0.05). Male adolescents also had a lower likelihood of low self-esteem compared to females (CPR = 0.96, p = 0.061). Similarly, those with tertiary education were less likely to report low self-esteem than those with no education (CPR = 0.95, p = 0.199). Limited social engagement, hydroxyurea use, school absence, and limitations in physical activity were associated with a higher likelihood of low self-esteem at bivariate analysis. These results are summarised in Table 2.

### Multi-variable factors associated with low self-esteem

Lower prevalence of low self-esteem was observed among adolescents not frequently absent from school (aPR = 0.94, 95% CI 0.90–0.99, P = 0.011) and those without limitations to physical activity (aPR = 0.94, 95% CI 0.90–0.98, P = 0.005), while limited social engagement was associated with higher prevalence (aPR = 1.16, 95% CI 1.03–1.31, P = 0.014)."The results are summarized in Table 3 below.

**Table 1. Baseline characteristics of adolescents with scd at MNRH.**

| Variable | Frequency | Percent (%) |
|---|---|---|
| **Age** | **N = 356** | |
| 10—14 | 235 | 66.0 |
| >14 | 121 | 34.0 |
| **Sex** | | |
| Male | 177 | 49.7 |
| Female | 179 | 50.3 |
| **Education** | | |
| Not completed primary | 207 | 58.2 |
| Completed Primary | 93 | 26.1 |
| Completed Secondary | 51 | 14.3 |
| Completed Tertiary | 5 | 1.4 |
| **Size of Family** | | |
| <6 | 145 | 40.7 |
| ≥6 | 211 | 59.3 |
| **Relative with SCD** | | |
| No | 103 | 28.9 |
| Yes | 253 | 71.1 |
| **Limited Social engagement** | | |
| No | 30 | 8.4 |
| Yes | 326 | 91.6 |
| **Hospitalisation in the last 12 months** | | |
| None | 123 | 34.6 |
| <4 | 165 | 46.3 |
| ≥4 | 68 | 19.1 |
| **Taking Hydroxyurea** | | |
| Yes | 317 | 89.1 |
| No | 39 | 10.9 |
| **In an intimate relationship** | | |
| Yes | 17 | 4.8 |
| No | 339 | 95.2 |
| **Frequently absent from school** **Days missed Median(IQR)** | **7(11)** | |
| No | 178 | 50.0 |
| Yes | 178 | 50.0 |
| **Limited physical activity** | | |
| Yes | 186 | 52.4 |
| No | 170 | 47.8 |

## Qualitative results

Twelve interviews were conducted—nine with adolescents (aged 10–18 years, 66% female) and three with health workers (pharmacist, medical officer, and nurse). Thematic analysis guided by the THRIVE framework, because it provides a needs-based approach to understanding adolescent health [11]. We identified coping mechanisms and factors influencing self-esteem among adolescents with sickle cell disease. The results are summarised in Table 4 below.

Table 2. Bivariate analysis of factors associated with low self-esteem among adolescents with SCD at MNRH.

| Variable | Self esteem | | Crude PR | 95% CI | P value |
|---|---|---|---|---|---|
| | Low(n=77) | Normal(n=279) | | | |
| **Age** | | | | | |
| 10—14 | 43(55.8) | 191(68.5) | 1.00 | | **0.05** |
| >14 | 34(44.2) | 88(31.5) | 0.95 | 0.90-1.00 | |
| **Sex** | | | | | |
| Male | 31(40.3) | 146(52.3) | 1.00 | | |
| Female | 46(59.7) | 133(47.7) | 0.96 | 0.91-1.00 | **0.061** |
| **Education** | | | | | |
| Not completed primary | 42(54.5) | 165(59.1) | 1.00 | | |
| Completed Primary | 18(23.4) | 74(26.5) | 1.00 | 0.95-1.06 | 0.885 |
| Completed Secondary | 15(19.5) | 37(13.3) | 0.95 | 0.88-1.03 | **0.199** |
| Completed Tertiary | 2(2.6) | 3(1.1) | 0.93 | 0.67-1.30 | 0.646 |
| **Employment** | | | | | |
| Employed | 11(14.3) | 25(9.0) | 1.00 | | |
| Not Employed | 66(85.7) | 254(91.0) | 1.06 | 0.96-1.17 | 0.233 |
| **Size of Family** | | | | | |
| <6 | 28(36.4) | 118(42.3) | 1.00 | | |
| ≥6 | 49(63.6) | 161(57.7) | 0.97 | 0.93-1.02 | 0.297 |
| **Relative with SCD** | | | | | |
| No | 17(22.1) | 86(30.8) | 1.00 | | |
| Yes | 60(77.9) | 193(69.2) | 0.98 | 0.90-1.06 | 0.562 |
| **Limited social engagement** | | | | | |
| No | 14(18.2) | 17(6.1) | 1.00 | | |
| Yes | 63(81.8) | 262(93.9) | 1.19 | 1.05-1.35 | **0.005** |
| **Hospitalisation in the last 12 months** | | | | | |
| None | 29(37.6) | 94(33.7) | 1.00 | | |
| <4 | 31(40.3) | 134(48.4) | 1.04 | 0.98-1.12 | 0.211 |
| ≥4 | 17(22.1) | 51(18.3) | 0.94 | 0.84-1.06 | 0.299 |
| **Taking Hydroxyurea** | | | | | |
| Yes | 64(83.1) | 253(91.7) | 1.00 | | |
| No | 13(16.9) | 26(9.3) | 0.92 | 0.84-1.01 | **0.092** |
| **In an intimate relationship** | | | | | |
| No | 73(94.8) | 266(95.3) | 1.00 | | |
| Yes | 4(5.2) | 13(4.6) | 0.99 | 0.88-1.11 | 0.851 |
| **Have family and peer support** | | | | | |
| No | 4(5.2) | 11(3.9) | 1.00 | | |
| Yes | 73(94.8) | 268(96.1) | 1.04 | 0.91-1.20 | 0.568 |
| **Frequently absent from school** | | | | | |
| No | 27(35.1) | 161(57.7) | 1.00 | | |
| Yes | 50(64.9) | 138(42.3) | 0.93 | 0.88-0.97 | **0.001** |
| **Limited physical activity** | | | | | |
| No | 24(31.2) | 146(52.3) | 1.00 | | |
| Yes | 53(69.8) | 133(47.7) | 0.92 | 0.88-0.96 | **0.001** |

**Table 3. Multivariable factors associated with low self-esteem among adolescents with SCD at MNRH.**

| Variable | Crude PR | Adjusted PR | 95% CI | P value |
|---|---|---|---|---|
| **Limited Social engagement** | | | | |
| **No** | 1.00 | 1.00 | | |
| **Yes** | 1.19 | 1.16 | 1.03-1.31 | **0.014** |
| | | | | |
| **Frequently absent from school** | | | | |
| **Frequently absent** | 1.00 | 1.00 | | |
| No absenteeism | 0.93 | 0.94 | 0.90-0.99 | **0.011** |
| **Limited Physical activity** | | | | |
| **Yes** | 1.00 | 1.00 | | |
| No | 0.92 | 0.94 | 0.90-0.98 | **0.005** |

**Table 4. Themes of coping mechanisms by the THRIVE framework.**

| Themes | Codes | categories |
|---|---|---|
| **Therapeutic interventions** | | |
| | Availability and Adherence to Hydroxyurea and Other Prescribed Medications | I make sure I take my medication |
| | | |
| **Habits and routines** | Nutrition and lifestyle (following a healthy diet, drinking plenty of water, and following clinic visits) | I take a lot of water |
| **Relational-social factors** | Support system (family, social, and peer support) Positive peer involvement. Diversionary activities like sports and games | My family is there for me whenever I need them |
| | Family support | My friends help me a lot |
| | Positive peer involvement | |
| **Valves and beliefs** | Belief in God and prayer Acceptance of their condition | I pray to God to help me live long |
| | | |
| **Emotional factors** | Hope and positive thinking Knowing that I am not alone | Other children with SCD are like me |
| | | |

## Availability and adherence to medication

Adolescents with normal self-esteem more commonly reported consistent adherence to medications such as hydroxyurea and folic acid compared to those with low self-esteem. Many described medication adherence as a source of reassurance and control over their condition.

"*I make sure I take my medication every day, and that's what makes me feel normal.*" **13-year-old female, Primary(P) 6.**

"*I used to fear taking my medication, but now I take it every day so that I don't get pain many times.*" — **16-year-old male, secondary(S) 2.**

Health workers also emphasized the psychological impact of drug availability. The availability of hydroxyurea at the SCC was seen as essential to both symptom control and emotional well-being.

"*They get depressed and stressed when they come, and you tell them that hydroxyurea is out of stock. So those that can buy it tend to have better self-esteem than those that can't.*" — **Hospital pharmacist**

"*There are those who can buy the hydroxyurea whenever it's out of stock at the SCC. They have fewer symptoms and worry less about their illness, giving them better self-esteem, for example, at school.*" — **Medical officer.**

### Healthy eating and hydration

Participants reported that maintaining healthy dietary and hydration habits helped them cope with the daily challenges of living with SCD. These behaviors were perceived to reduce the frequency and severity of pain episodes.

"*I drink plenty of water at home and also while at school. I always ensure I have water in my bottle.*" — **16-year-old female, S.2**

"*I drink juice and eat fruits, which helps to prevent the pain.*" —**18-year-old female, S.5**

### Engagement in social activities and hobbies

Adolescents with both low and normal self-esteem reported engaging in social and recreational activities to distract from the emotional burden of SCD. Participation in school events and personal hobbies contributed to a sense of normalcy and inclusion.

"*I try to participate in school activities, but the teachers can only allow me to participate in singing.*" — **16-year-old male, S.2**

"*Teachers allow me to participate in all sports. This makes me feel like the other students.*" — **12-year-old female, P.6**

"*I keep busy by reading books, watching TV, and cartoons, and this stops me from thinking about the illness.*" — **11-year-old male, P.4**

### Spiritual support

Spirituality and prayer emerged as important sources of strength and hope for several participants. Many adolescents expressed a reliance on faith as a coping mechanism.

"Praying to God every time I go to church and even at home gives me strength." — **11-year-old female, P.4**

"*I pray to God to give me a better and longer life. I hear that children with SCD don't live beyond 18 years.*" — ***10-year-old male, P.4***

### Social support from peers and school staff

Having peers with SCD or receiving empathy from adults familiar with the condition helped adolescents feel less isolated and more understood.

"*At home, I'm the only one with SCD, but at school, there are students with SCD, so I'm not alone.*" — **18-year-old female, S.5**

"*The school nurse has a daughter with SCD, and when she talks to me about her daughter, I feel better.*" — **16-year-old female, S.2**

### Caregiver support

Health workers observed that adolescents accompanied by their mothers to clinic visits tended to show better coping and emotional outcomes. Maternal involvement was associated with better pain management and higher self-esteem.

"*The children that come to the clinic with their mothers have better pain control than the others, and this gives them better self-esteem.*" — ***SCC nurse.***

## 4. Discussion

### Prevalence of low self-esteem

The prevalence of low self-esteem (LSE) was 21.6%. This lower prevalence may be attributed to strong family and peer support, and widespread use of hydroxyurea among participants. Hydroxyurea improves quality of life by reducing pain episodes and hospitalizations and by minimizing physical manifestations of SCD, such as jaundice and stunted growth, thereby potentially enhancing self-esteem [3].

A similar prevalence was observed in a study conducted by Gidi (2021) in Ethiopia among university students aged 17–24 years, which found an LSE prevalence of 19% [12]. However, our findings contrast with studies by Engoba et al. in Brazzaville and Seigel at el in the U.S., which reported LSE rates of 76.1% and 75% among adolescents with SCD, respectively [13,14]. The discrepancies may be due to differences in study settings, sample characteristics, and psychosocial support systems.

### Factors associated with low self-esteem

Key factors significantly associated with LSE in this study included limited social involvement, poor physical activity, and school absenteeism. Demographic characteristics such as age, sex, and education level were not associated with LSE, consistent with other findings [15–17].

**Limited social engagement.** Limited social engagement was associated with a higher prevalence of LSE. Adolescents with Social isolation often lead to stigma, reduced peer interaction, and a lack of validation, which adversely affect self-esteem. Studies by Alvin (2003) and Forrester et al. (2015) support these findings, showing that adolescents with SCD tend to isolate themselves and that those involved in social and cultural activities develop more positive self-concepts [7,16,18]. Similar observations have been reported in Nigeria and other sub-Saharan contexts [15,17].

**Limited physical activity.** Lower prevalence of low self-esteem was observed among adolescents without limitations to physical activity. Fear of triggering pain crises discourages physical exertion, thereby limiting self-expression and peer interaction. This not only contributes to poor body image but also hampers self-efficacy and self-worth [7,16,19]. A study in Switzerland reported similar findings, where adolescents with SCD who had reduced physical activity also had a diminished sense of self-concept [20]. Health care providers, parents, and teachers should encourage adolescents with SCD to participate in safe and appropriate physical and social activities, which can foster resilience and promote healthy self-esteem

**School absenteeism.** A lower prevalence of low self-esteem was observed among adolescents not frequently absent from school. Common reasons included pain crises and overprotective behavior by caregivers. Absenteeism disrupts

academic progress, social interaction, and participation in extracurricular activities, contributing to feelings of disconnection, inadequacy, and reduced self-esteem [4,21]. Similar associations were reported in studies conducted in Brazzaville and by Broome et al., where frequent hospitalizations and poor school participation were linked to poor psychosocial outcomes [14,22]. In addition, collaborative efforts should be undertaken to minimize school absenteeism by providing necessary support. Maintaining regular school attendance is crucial for enhancing self-esteem and promoting overall development.

## Coping mechanisms

Adolescents used a range of coping mechanisms across six domains:

**Therapeutic interventions.** Adolescents reported that adherence to hydroxyurea, folic acid, and antimalarial prophylaxis (fansidar) reduced hospital visits and pain, allowing participation in school and social activities, ultimately improving their self-esteem. This aligns with findings by Kambasu et al. and Sarah et al., who demonstrated that adherence to medication improves health outcomes and self-esteem [3,23].

**Habits and routine.** Maintaining medical appointments and following dietary guidelines were reported as coping strategies. These routines helped reduce complications, leading to improved well-being and self-esteem [3,7,22].

**Relational and social factors.** Strong family and peer support were recurring themes. Adolescents who felt accepted and supported reported less stress and higher self-worth. This aligns with previous studies that highlighted the positive role of social networks in psychological adjustment among adolescents with SCD [7,15,24]. Participation in extracurricular and leisure activities helped distract from illness-related stress. However, barriers to sports participation due to perceived health risks were still reported [7,18].

**Values and beliefs.** Many adolescents expressed a strong belief in God and reported that prayer helped them cope with pain and stress. They also expressed optimism about managing their condition if they adhered to medical advice. These findings are consistent with studies from Nigeria, where spiritual beliefs provided comfort and hope [15,24,25].

**Emotional factors.** Knowing others with SCD gave adolescents a sense of belonging and reduced feelings of isolation. Sharing experiences with peers fostered emotional resilience and improved self-esteem [15,26,27]. Most participants had hope for better health and did not exhibit self-blame, which promoted a sense of control and psychological well-being [5].

## 4.1 Study limitations

Its cross-sectional design prevents establishing causal relationships. The Rosenberg Self-Esteem Scale (RSES) was not piloted in this setting, and although widely used, its cultural validity in Uganda is uncertain, introducing possible measurement bias. Self-reported responses may also have been affected by social desirability. Some misclassification may have occurred for variables such as schooling level and activity limitations, as these were based on participant recall. The clinic-based sample, drawn from a national referral hospital, may limit generalizability to adolescents in other settings. A conservative cut-off of ≤ 15 used to define LSE may have underestimated its prevalence. We acknowledge the potential developmental limitations of applying the RSES to participants younger than 12 years in this study. Missing data were minimal and handled using complete-case analysis, though some bias cannot be entirely excluded.

## 5. Conclusion

The prevalence of low self-esteem among adolescents with sickle cell disease (SCD) was 21.6%. Physical activity and social involvement were found to play a vital role in shaping self-esteem among these adolescents, while school absenteeism emerged as a key factor contributing to reduced self-worth.

Effective coping strategies such as prayer, adherence to medications and clinic appointments, participation in diversionary activities (e.g., sports and television), social support, and peer networks can help adolescents with SCD maintain better self-esteem and mental well-being.

## Author contributions

**Conceptualization:** Janipher Nakabuye, Sarah Kiguli, Deogratias Munube.

**Formal analysis:** Wani Muzeyi.

**Funding acquisition:** Sarah Kiguli.

**Investigation:** Janipher Nakabuye.

**Methodology:** Janipher Nakabuye.

**Software:** Wani Muzeyi.

**Supervision:** Annet Nakirulu, Deogratias Munube, Grace Ndeezi.

**Writing – original draft:** Janipher Nakabuye, Wani Muzeyi.

**Writing – review & editing:** Janipher Nakabuye, Wani Muzeyi, Grace Ndeezi.

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
