## [Decision Letter · Decision Letter 0]

29 Oct 2025

Low Self-Esteem in Adolescents with Sickle Cell Disease: Determinants and Coping Mechanisms at a tertiary hospital, Kampala, Uganda; A mixed methods study.

PLOS ONE

Dear Dr. Nakabuye,

Thank you for submitting your manuscript to PLOS ONE. After careful consideration, we feel that it has merit but does not fully meet PLOS ONE’s publication criteria as it currently stands. Therefore, we invite you to submit a revised version of the manuscript that addresses the points raised during the review process.

We look forward to receiving your revised manuscript.

Kind regards,

Ghada O. Wassif, M.D.

Academic Editor

PLOS ONE

2. We note that there is identifying data in the Supporting Information file  <Supporting_files.rar>. Due to the inclusion of these potentially identifying data, we have removed this file from your file inventory. Prior to sharing human research participant data, authors should consult with an ethics committee to ensure data are shared in accordance with participant consent and all applicable local laws.

-Location data

3. We note you have included a table to which you do not refer in the text of your manuscript. Please ensure that you refer to Table 4 in your text; if accepted, production will need this reference to link the reader to the Table.

4. Please include captions for your Supporting Information files at the end of your manuscript, and update any in-text citations to match accordingly. Please see our Supporting Information guidelines for more information: http://journals.plos.org/plosone/s/supporting-information .

Additional Editor Comments:

Subject: Revision Request for Manuscript ID: PONE-D-25-41712

Dear Dr., Nakabuye,

Thank you for submitting your manuscript to PLOS ONE. Following a thorough review by the editorial board and external peer reviewers, we are pleased to inform you that your manuscript is considered potentially suitable for publication pending a detailed revision.

To proceed with the editorial process, we require that you carefully address all comments raised by the reviewers and the Associate Editor. Please note the following revision instructions:

Revision Instructions

1. Prepare a Point-by-Point Response Letter:

Please respond individually to each reviewer and editor comment. Indicate how each concern has been addressed and where changes appear in the revised manuscript.

2. Highlight Revisions in the Manuscript:

o Use color-coded highlights in the revised manuscript to indicate responses to each source:

Reviewer 1 – Blue

Reviewer 2 – Green

Reviewer 3 (if applicable) – Purple

Associate Editor – Yellow

o If a section was added or rewritten entirely, highlight the new text in the designated color.

o Do not use track changes; color coding only is preferred to ensure readability.

3. Ensure Consistency with Journal Guidelines:

Please make sure your formatting, citations, tables, figures, and declarations align with PLOS ONE requirements.

We appreciate the significance of your work and the constructive dialogue it has opened. We look forward to receiving your revised submission within 30 days. Should you require an extension, please contact us at your earliest convenience.

Thank you for your continued contribution to the journal.

Sincerely,

Ghada O. Wassif

Academic Editor

PLOS ONE

ghadawasif@med.asu.edu.eg

Academic Editor Comments

1) Title, authors, affiliations, and front matter

• Title (remove place of study): The checklist asks to delete the place of the study from the title. Recommend:

o Proposed title: “Low Self-Esteem in Adolescents with Sickle Cell Disease: Determinants and Coping Mechanisms — A Mixed-Methods Study.”

• Affiliation formatting & consistency: Use only “Department, University, City” (capitalize proper nouns; singular/plural consistently). E.g., “Department of Pediatrics and Child Health, Makerere University College of Health Sciences, Kampala, Uganda.” Ensure the two affiliations follow the same pattern. Fix capitalization (“Makerere University,” not “Makerere university”).

• Name inconsistency: “Grace Ndeezi” vs “Grace Ndezi” (one ‘e’ missing in the author line). Standardize to the correct spelling throughout.

• Corresponding author details: Email and phone are present; keep only in the author block (do not repeat elsewhere).

• Numbering of sections: Use journal-consistent numbering (e.g., 1. Introduction; 2. Materials and Methods; 3. Results; 4. Discussion; 4.1 Limitations; 5. Conclusion).

2) Abstract

• Structure: Generally appropriate (Background, Methods, Results, Conclusion).

• Sample size placement: Good—sample size is in Methods (n=356).

• Effect directions & precision: Clarify directionality. As coded in the model, “No” school absence and “No” physical limitation are protective (PRs < 1), while limited social engagement increases prevalence (PR > 1). Rewrite to avoid ambiguity like “included school absenteeism” without direction. Suggest:

o “Lower prevalence of low self-esteem was observed among adolescents not frequently absent from school (aPR = 0.94, 95% CI 0.90–0.99) and those without limitations to physical activity (aPR = 0.94, 95% CI 0.90–0.98), while limited social engagement was associated with higher prevalence (aPR = 1.16, 95% CI 1.03–1.31).”

• Statistics in abstract: Keep measures (PRs, CIs, p-values) but don’t mention software names here (that belongs in Methods).

3) Introduction

• Currency & concision: Reasonable framing; tighten historical/global figures to 1–2 citations and remove duplication.

• Terminology consistency: Use THRIVE (all caps) consistently (appears as “Thrive/THRIVE” in different places).

• Aim statement: Place the aim at the end of the Introduction as a final sentence, not as a separate paragraph.

4) Materials and Methods

• Design & setting: Clear mixed-methods, sequential explanatory; dates and clinic context included. Good.

• Sample size inconsistency: Text states “Three hundred and fifty-seven (356)”—fix to 356 everywhere (and briefly explain any exclusions if applicable).

• Sampling & response rate: Describe sampling approach (consecutive? convenience? systematic?) and report response rate (invited vs enrolled), per checklist.

• Measurement validity & reliability: Add validity/translation information for Rosenberg Self-Esteem Scale (RSES) in this population (e.g., cultural/linguistic adaptation, Cronbach’s alpha in your sample). This is currently missing and required by the checklist.

• Variable definitions: You defined low self-esteem as RSES < 15—state rationale with citation.

• Statistical analysis: You stated modified Poisson with robust SEs and p≤0.20 for entry—good. Add specifics: which covariates entered a priori vs. data-driven; handling of missing data; checks for multicollinearity; model fit diagnostics (if any).

• Qualitative methods: Nice detail (IDIs/KIIs, language, transcription, coding, saturation). Add reference or short description of the THRIVE framework and clarify why it guided analysis.

• Ethics/consent: Approval number is present; include approval date and specify written consent/assent clearly (parents/guardians vs 18–19y participants) inside Methods (not only in front matter), per checklist.

5) Results

• Participant characteristics:

o Wording check: “Most (58.5%) had no formal education.” For adolescents 10–19, this is unusual; likely means “no secondary education” or “not beyond primary.” Ensure accuracy of the label and interpretation. The table shows “None, Primary, Secondary, Tertiary”—verify data coding, then rewrite text to match actual schooling stages.

• Tables: numbering, captions, and footnotes:

o Tables jump (1, 3, 4, 7). Ensure a continuous sequence (Table 1, 2, 3, …).

o Title format per checklist: include participants, place, and time (e.g., “Baseline characteristics of adolescents with SCD attending MNRH SCC, Kampala, Uganda (Jan 8–Mar 26, 2024), N=356”). No full stop at the end of table titles.

o Standardize column headers to “No. (%)” and remove “%” from cell entries.

o Keep totals to exactly 100% (round carefully; add footnote if rounding causes ±0.1%).

o Use asterisks solely for significance; add footnotes to explain symbols.

o Per checklist, report p-values only (omit test statistics like χ² in tables).

• Bivariate text accuracy: The narrative says “and so were taking hydroxyurea” among factors associated with LSE, but the table shows PR = 0.92, p = 0.092 (not significant). Remove that claim or qualify it as non-significant.

• Multivariable model wording: Present contrasts clearly (what is reference category; write “Compared with those frequently absent from school, adolescents not frequently absent had lower prevalence of LSE (aPR…)).” Ensure the narrative matches the direction of the coefficients you present.

• Figure(s): If any figures exist in the source files, ensure legends are complete (participants, place, time). If none, consider a single adjusted PR forest plot for the three key variables to improve clarity (optional but helpful).

6) Qualitative results

• Framework & themes: Clear five-domain synthesis consistent with THRIVE; quotes are appropriate.

• Attribution: Keep anonymized age/sex/grade labels consistent; ensure all quotes are translated and indicated as such when not originally in English.

7) Discussion

• Start with key findings (no mini-introduction).

• Avoid repetition of numerical results. Interpret directions: social limitation ↑LSE prevalence; school presence and physical ability ↓LSE prevalence.

• Literature consistency:

o Fix author name inconsistencies used in-text: you cite “Forrestal” in prose but the reference lists “Forrester AB”—standardize.

o “Siegel/Seigel” spellings differ between text and references—standardize.

• Strengths & limitations: Move Limitations to its own subsection (see next section) and avoid exaggeration in Strengths.

8) Limitations (separate subsection “4.1 Limitations”)

Create a brief, explicit subsection that discusses:

• Cross-sectional design (no causal inference).

• Potential measurement bias (RSES cultural validity; social desirability).

• Possible misclassification of schooling level and activity limitations.

• Clinic-based sample (generalizability).

• Missing data handling (if any).

This is specifically required by the checklist.

9) Conclusion

• Keep it brief and answer the aim directly, without references. Add one-sentence practical implication (e.g., integrate psychosocial screening and activity-friendly guidance in clinic SOPs).

10) Ethics, data availability, funding, competing interests

• Ethics statement: Add the approval date and keep the approval code; ensure this exact text also appears within Methods.

• Data availability: Statement says “All relevant data are within the manuscript and its Supporting Information files.” Ensure the Supporting files link(s) are live at submission and mention file names/types in the statement.

• Funding: Provide funder details per PLOS format (initials, grant number, funder name, URL, and the no-role statement). Your draft text is close; just ensure initials are included for recipients and add the funder URL(s).

• Competing interests: “The authors have declared that no competing interests exist.” Fine.

11) Language and style

• Voice: Checklist prefers first person over heavy passive constructions (e.g., “We conducted… We used…”).

• Consistency: Use “adolescents with SCD” throughout; avoid mixing “children/teenagers” unless defined.

• Spelling/typos: Fix “Thrive”→“THRIVE”; “college of health sciences”→“College of Health Sciences”; “Mulago national referral hospital”→“Mulago National Referral Hospital”; “Ndeezi/Ndezi” consistency.

12) References (Vancouver, with DOIs/URLs)

• Vancouver compliance: Re-format all references strictly to Vancouver style with abbreviated journal titles.

• Correct author names & titles: Align in-text citations with the reference list (Forrester vs Forrestal; Siegel vs Seigel; Pediatric Blood & Cancer capitalization).

• Add DOIs for journal articles wherever available; for websites include URL and access date.

• Count: ~40 is typical for an original article; your count is within range but finalize after deduplication and corrections.

Reviewer 1

The study addresses a timely and important gap: mental health, specifically self-esteem, among adolescents with sickle cell disease (SCD) in a low-resource setting. The mixed-methods design is appropriate, and the integration of qualitative insights (example, coping strategies like prayer, peer support, and medication adherence) enriches the quantitative findings.

Here are some major methodological concerns that undermine technical soundness:

• Unconventional RSES cutoff: The authors define low self-esteem as a Rosenberg Self-Esteem Scale (RSES) score <15 (lines 30–31, 102–103). This is atypical. The standard RSES ranges from 0–30, and most studies (including in African contexts) use ≤25 or ≤20 to indicate low self-esteem. A cutoff of <15 suggests only the most severely affected are classified as having low self-esteem, which likely underestimates prevalence and may explain why the reported prevalence (21.6%) is lower than in comparable studies (e.g., 75–76% in U.S. and Congo; lines 260–262). The authors do not justify this threshold, nor cite validation studies supporting its use in Ugandan adolescents. This is a critical flaw that affects all downstream analyses and conclusions.

• Key predictors like “limited physical activity” and “limited social engagement” (lines 170, Table 1) are not operationally defined. How were these assessed? Were standardized tools used, or was this based on a single-item self-report? Without clarity, reproducibility and validity are compromised.

• In Table 4, the reference group for school absenteeism is “Yes” (frequently absent), and the adjusted PR for “No” is 0.94. While statistically correct, this framing is counterintuitive and risks misinterpretation. It would be clearer to code “No absenteeism” as the reference so that PR >1 reflects increased risk. As written, readers may mistakenly infer that absenteeism is protective.

• Age range concerns: Including 10-year-olds (line 94) in an “adolescent” study using the RSES, a scale validated primarily for ages 12+, raises developmental validity questions. Young children may interpret items differently, potentially biasing their scores.

The use of modified Poisson regression to estimate prevalence ratios is appropriate for common outcomes (low self-esteem prevalence >10%). However:

• The manuscript states that variables with p ≤ 0.2 in bivariate analysis were eligible for multivariable modeling (lines 132–133). Yet, several such variables (e.g., hydroxyurea use, p = 0.092; hospitalization, p = 0.211) were excluded from the final model without justification (Table 3 vs. Table 4). This selective inclusion weakens transparency.

• No model diagnostics (e.g., goodness-of-fit, multicollinearity checks) are reported. Given the small number of covariates, this is a missed opportunity to strengthen rigor.

• Dichotomizing the RSES discards valuable information and reduces statistical power. A sensitivity analysis using the continuous RSES score would strengthen the findings.

Strengths

• The qualitative component is rich and contextually grounded, offering actionable insights (e.g., the role of faith, peer awareness, caregiver support).

• The focus on modifiable factors (school attendance, physical activity, social engagement) provides a foundation for interventions.

This is a valuable and socially relevant study that sheds light on an under-addressed aspect of SCD care in Uganda. With methodological clarifications, particularly regarding the RSES cutoff and variable definitions, the manuscript could make a strong contribution to the literature. I recommend a major revision before acceptance.

Reviewer 2

Overall, this is a well-written paper that handles an important discussion surrounding self-esteem in adolescents with sickle cell disease thoughtfully and skillfully. I have a few suggestions to improve the paper, as listed below:

- I would recommend re-reviewing the paper to ensure that all sentences are complete (e.g. line 70, lines 94-95).

- Table 1 has a category for Limited Social Engagement (91.6% reported "Yes"), while also separately reporting "Have family and peer support" (95.8% reported "Yes"). Would you please expand upon the differences between these two questions? This will also need explained further in the paper when the table is discussed.

- Line 186 mentions the THRIVE framework, but I did not see this explained elsewhere in the paper. Could you please define this/provide a reference for this, as it is not common knowledge?

- I would suggest adding a different heading prior to your section listing direct quotes from study participants to better frame that section of the paper.

Reviewer's Responses to Questions

**Comments to the Author**

1. Is the manuscript technically sound, and do the data support the conclusions?

Reviewer #1: Yes

Reviewer #2: Partly

2. Has the statistical analysis been performed appropriately and rigorously?

Reviewer #1: Yes

Reviewer #2: Yes

3. Have the authors made all data underlying the findings in their manuscript fully available?

Reviewer #1: Yes

Reviewer #2: Yes

4. Is the manuscript presented in an intelligible fashion and written in standard English?

Reviewer #1: Yes

Reviewer #2: Yes

Reviewer #1: Overall, this is a well-written paper that handles an important discussion surrounding self-esteem in adolescents with sickle cell disease thoughtfully and skillfully. I have a few suggestions to improve the paper, as listed below:

- I would recommend re-reviewing the paper to ensure that all sentences are complete (e.g. line 70, lines 94-95).

- Table 1 has a category for Limited Social Engagement (91.6% reported "Yes"), while also separately reporting "Have family and peer support" (95.8% reported "Yes"). Would you please expand upon the differences between these two questions? This will also need explained further in the paper when the table is discussed.

- Line 186 mentions the THRIVE framework, but I did not see this explained elsewhere in the paper. Could you please define this/provide a reference for this, as it is not common knowledge?

- I would suggest adding a different heading prior to your section listing direct quotes from study participants to better frame that section of the paper.

Reviewer #2: The study addresses a timely and important gap: mental health, specifically self-esteem, among adolescents with sickle cell disease (SCD) in a low-resource setting. The mixed-methods design is appropriate, and the integration of qualitative insights (example, coping strategies like prayer, peer support, and medication adherence) enriches the quantitative findings.

Here are some major methodological concerns that undermine technical soundness:

• Unconventional RSES cutoff: The authors define low self-esteem as a Rosenberg Self-Esteem Scale (RSES) score <15 (lines 30–31, 102–103). This is atypical. The standard RSES ranges from 0–30, and most studies (including in African contexts) use ≤25 or ≤20 to indicate low self-esteem. A cutoff of <15 suggests only the most severely affected are classified as having low self-esteem, which likely underestimates prevalence and may explain why the reported prevalence (21.6%) is lower than in comparable studies (e.g., 75–76% in U.S. and Congo; lines 260–262). The authors do not justify this threshold, nor cite validation studies supporting its use in Ugandan adolescents. This is a critical flaw that affects all downstream analyses and conclusions.

• Key predictors like “limited physical activity” and “limited social engagement” (lines 170, Table 1) are not operationally defined. How were these assessed? Were standardized tools used, or was this based on a single-item self-report? Without clarity, reproducibility and validity are compromised.

• In Table 4, the reference group for school absenteeism is “Yes” (frequently absent), and the adjusted PR for “No” is 0.94. While statistically correct, this framing is counterintuitive and risks misinterpretation. It would be clearer to code “No absenteeism” as the reference so that PR >1 reflects increased risk. As written, readers may mistakenly infer that absenteeism is protective.

• Age range concerns: Including 10-year-olds (line 94) in an “adolescent” study using the RSES, a scale validated primarily for ages 12+, raises developmental validity questions. Young children may interpret items differently, potentially biasing their scores.

The use of modified Poisson regression to estimate prevalence ratios is appropriate for common outcomes (low self-esteem prevalence >10%). However:

• The manuscript states that variables with p ≤ 0.2 in bivariate analysis were eligible for multivariable modeling (lines 132–133). Yet, several such variables (e.g., hydroxyurea use, p = 0.092; hospitalization, p = 0.211) were excluded from the final model without justification (Table 3 vs. Table 4). This selective inclusion weakens transparency.

• No model diagnostics (e.g., goodness-of-fit, multicollinearity checks) are reported. Given the small number of covariates, this is a missed opportunity to strengthen rigor.

• Dichotomizing the RSES discards valuable information and reduces statistical power. A sensitivity analysis using the continuous RSES score would strengthen the findings.

Strengths

• The qualitative component is rich and contextually grounded, offering actionable insights (e.g., the role of faith, peer awareness, caregiver support).

• The focus on modifiable factors (school attendance, physical activity, social engagement) provides a foundation for interventions

This is a valuable and socially relevant study that sheds light on an under-addressed aspect of SCD care in Uganda. With methodological clarifications, particularly regarding the RSES cutoff and variable definitions, the manuscript could make a strong contribution to the literature. I recommend a major revision before acceptance.

**Do you want your identity to be public for this peer review?**  For information about this choice, including consent withdrawal, please see our Privacy Policy

Reviewer #1: No

Reviewer #2: No

---

## [Author Response · Author response to Decision Letter 1]

15 Dec 2025

A rebuttal letter that responds to each point raised by academic editor and reviewer(s), a marked up copy of the manuscript that highlights changes made to the original manuscript and an un marked version of my revised paper have been uploaded as guided.

Quantitative data submitted has been deidentified to protect the paticipants.

---

## [Decision Letter · Decision Letter 1]

9 Feb 2026

Dear Dr. Nakabuye,

Thank you for submitting your manuscript to PLOS ONE. After careful consideration, we feel that it has merit but does not fully meet PLOS ONE’s publication criteria as it currently stands. Therefore, we invite you to submit a revised version of the manuscript that addresses the points raised during the review process.

We look forward to receiving your revised manuscript.

Kind regards,

Tomasz W. Kaminski

Academic Editor

PLOS One

Journal Requirements:

Additional Editor Comments:

Dear Authors,

Thank you for submitting the revised version of your manuscript. The reviewers’ comments have been adequately addressed, and the overall scientific content is sound. The manuscript is close to being acceptable for publication.

Before a final decision can be made, please address the following minor editorial and technical issues identified during final assessment. These points do not affect the study’s conclusions but require correction for clarity and consistency:

- Correction of a typographical error in the reported p-value for limited social engagement, ensuring consistency between the Abstract, Results text, and Table 3. In the abstract there is "while limited social engagement was associated with higher prevalence (aPR = 1.16, 95% CI 1.03–1.31, P=1.16)" - P=1.16 is not possible.

- Verification and harmonization of reported prevalence ratios and p-values across the Abstract, Results section, and corresponding tables - this must be carefully addressed during this minor revision stage.

- Minor clarification of statistical terminology and wording in the Methods section, particularly regarding covariate selection and multivariable modeling:

"Covariate selection was data driven." - too vague and doesn't allow reproducibility

"Multicollinearity was assessed using variance inflation factors (VIF), with no variable exceeding a VIF of 10 was retained in the model." - Grammatically incorrect and ambiguous logic

"The majority of participants (65%) were aged 10–14 years, and just over half were female (50.4%." - missing closing paranthesis

"Limited social engagement was associated with a higher likelihood of low self-esteem (CPR = 1.19, p = 0.005) and so was taking hydroxyurea…" - "and so was” is informal and unclear and hydroxyurea association is counterintuitive

"While this is high, it is lower than reported in similar studies" - Vague comparator (“similar studies”)

- Correction of grammatical errors and incomplete sentences in the Results and Discussion sections

- Correction of typographical errors in table titles and headings (e.g., wording related to coping mechanisms)

- Final consistency checks between the narrative descriptions of results and the data presented in tables

Please submit a revised version addressing the above points. Once these minor issues are resolved, the manuscript should be suitable for acceptance.

Kind regards,

Tomasz W Kaminski

Reviewers' comments:

Reviewer's Responses to Questions

**Comments to the Author**

Reviewer #2: All comments have been addressed

2. Is the manuscript technically sound, and do the data support the conclusions?

Reviewer #2: Yes

3. Has the statistical analysis been performed appropriately and rigorously?

Reviewer #2: Yes

4. Have the authors made all data underlying the findings in their manuscript fully available?

Reviewer #2: Yes

5. Is the manuscript presented in an intelligible fashion and written in standard English?

Reviewer #2: Yes

Reviewer #2: (No Response)

**Do you want your identity to be public for this peer review?** For information about this choice, including consent withdrawal, please see our Privacy Policy

Reviewer #2: No

---

## [Author Response · Author response to Decision Letter 2]

24 Feb 2026

Response to reviewers letter , a marked up copy of the manuscript that highlights changes made to the original manuscript and an un marked version of my revised paper have been uploaded as guided.

---

## [Editor Report · Decision Letter 2]

27 Feb 2026

Low Self-Esteem in Adolescents with Sickle Cell Disease: Determinants and Coping Mechanisms; A mixed methods study.

PONE-D-25-41712R2

Dear Dr. Nakabuye,

We’re pleased to inform you that your manuscript has been judged scientifically suitable for publication and will be formally accepted for publication once it meets all outstanding technical requirements.

Kind regards,

Tomasz W. Kaminski

Academic Editor

PLOS One

Reviewers' comments:

N/A

---

## [Editor Report · Acceptance letter]

PONE-D-25-41712R2

PLOS One

Dear Dr. Nakabuye,

I'm pleased to inform you that your manuscript has been deemed suitable for publication in PLOS One. Congratulations! Your manuscript is now being handed over to our production team.

Kind regards,

on behalf of

Dr. Tomasz W. Kaminski

Academic Editor

PLOS One